# The Influence of Formulation Components and Environmental Humidity on Spray-Dried Phage Powders for Treatment of Respiratory Infections Caused by *Acinetobacter baumannii*

**DOI:** 10.3390/pharmaceutics13081162

**Published:** 2021-07-28

**Authors:** Wei Yan, Ruide He, Xiaojiao Tang, Bin Tian, Yannan Liu, Yigang Tong, Kenneth K. W. To, Sharon S. Y. Leung

**Affiliations:** 1School of Pharmacy, Faculty of Medicine, The Chinese University of Hong Kong, Hong Kong Shatin, NewTerritories, Hong Kong, China; 1155114125@link.cuhk.edu.hk (W.Y.); kennethto@cuhk.edu.hk (K.K.W.T.); 2Livzon Pharmaceutical Group Co., Ltd., Zhu Hai 519090, China; heruide2021@163.com (R.H.); tangxiaojiaoxfx@163.com (X.T.); 3Department of Pharmaceutical Sciences, School of Food and Biological Engineering, Shanxi University of Science and Technology, Weiyang University Park, Xi’an 710021, China; tianbin@sust.edu.cn; 4Emergency Medicine Clinical Research Center, Beijing Chao-Yang Hospital, Capital Medical University, Beijing 100069, China; yannan_liu@foxmail.com; 5Beijing Advanced Innovation Center for Soft Matter Science and Engineering (BAIC-SM), College of Life Science and Technology, Beijing University of Chemical Technology, Beijing 100029, China; tongyigang@mail.buct.edu.cn

**Keywords:** phage therapy, spray drying, bacterial lung infections, environmental humidity, aerosol performance, inhalation

## Abstract

The feasibility of using respirable bacteriophage (phage) powder to treat lung infections has been demonstrated in animal models and clinical studies. This work investigated the influence of formulation compositions and excipient concentrations on the aerosol performance and storage stability of phage powder. An anti-*Acinetobacter baumannii* phage vB_AbaM-IME-AB406 was incorporated into dry powders consisting of trehalose, mannitol and L-leucine for the first time. The phage stability upon the spray-drying process, room temperature storage and powder dispersion under different humidity conditions were assessed. In general, powders prepared with higher mannitol content (40% of the total solids) showed a lower degree of particle merging and no sense of stickiness during sample handling. These formulations also provided better storage stability of phage with no further titer loss after 1 month and <1 log titer loss in 6 months at high excipient concentration. Mannitol improved the dispersibility of phage powders, but the in vitro lung dose dropped sharply after exposure to high-humidity condition (65% RH) for formulations with 20% mannitol. While previously collected knowledge on phage powder preparation could be largely extended to formulate *A. baumannii* phage into inhalable dry powders, the environmental humidity may have great impacts on the stability and dispersion of phage; therefore, specific attention is required when optimizing phage powder formulations for global distribution.

## 1. Introduction

With the overuse and misuse of antibiotics, antimicrobial resistance (AMR) has become one of the greatest threats to human health worldwide [1]. To guide research efforts aiming to develop new treatment strategies, the World Health Organization (WHO) published the first global priority pathogen list in 2017 [2]. Among all identified bacteria, *Acinetobacter baumannii* has been ranked as the number one critical-priority pathogen, emphasizing the extent of the threats posed to human health and the urgency for new treatment strategies. In fact, *A. baumannii* is generally regarded as a low-virulence pathogen. However, it has such an exceptional ability to acquire resistance to antibiotics that the resistance rates of multidrug-resistant (MDR), extensively drug-resistant (XDR), and pandrug-resistant (PDR) *A. baumannii* were 50.2%, 28.5%, and 14.0%, respectively, presenting a significant clinical challenge in choosing appropriate treatment regimens [3]. Last-line antibiotics, such as polymyxins and carbapenems, have been increasingly used as the only therapeutic option for life-threatening infections [4], but outbreaks of PDR *A. baumannii* have been increasing reported worldwide with a mortality rate of 26.0–55.7% [5].

Bacteriophage (phage) therapy is now being re-introduced in Western countries [6] and has been listed as one of the top alternatives to address the AMR threat [3]. The safety and efficacy of phage in treating patients against drug-resistant *A. baumannii* have been demonstrated in multiple cases of life-saving therapeutic use [7,8,9]. With pneumonia being one of the most frequent clinical manifestations of *A. baumannii* infection, increasing attention has been given to the investigation of the effectiveness of *A. baumannii* phage in treating lung infections [10,11,12,13,14]. Although nebulization has been the method of choice for phage delivery to the lung in the clinical setting, dry powder formulations are preferred to liquid formulations in terms of storage, transportation and administration [15].

Recently, spray drying has been demonstrated as a promising single-step process in producing inhalable phage dry powder formulations using low-cost excipients, such as trehalose, lactose, mannitol and leucine, showing a sufficiently long storage stability (≤1 log titer loss in 12 months) under refrigerated or room temperatures in low-humidity conditions (<20% RH) [16,17,18,19,20,21]. The efficacy of intratracheally administered spray-dried phage powder against lung infection caused by *Pseudomonas aeruginosa* has been demonstrated in a murine model [22]. To the best of our knowledge, there are no reports on formulating *Acinetobacter* phage into inhalable powders. As previous reports highlighted that the detrimental effects to phage upon the production process and storage conditions are phage dependent and excipient context specific [18,20,23], current knowledge obtained for *Pseudomonas* and other phages might not be directly applicable for *A. baumannii* phage. In the present study, our primary objective is to investigate the feasibility of extending the collected knowledge to produce inhalable *Acinetobacter* phage powders for inhaled therapy. A two-factor three-level factorial design [24] was used to evaluate the impacts of the two identified formulation factors, formulation compositions and total solid content, on the phage stability and in vitro aerosol performance of produced powders, using an *A. baumannii* phage vB_AbaM-IME-AB406 (AB406 in short) as a model phage.

As phages were usually stabilized with amorphous sugar in the powder form, their handling and storage in low-humidity conditions (RH < 20%) are required to minimize the occurrence of recrystallization. However, excessive environmental moisture could be relevant in patients’ homes or healthcare settings, especially in areas with subtropical climates such as Hong Kong and southern USA, where the average RH ≥ 65% all year round [25,26,27]. To estimate the impacts of high humidity on the administration of phage powders, we measured the in vitro aerosolization performance of phage powders after incubating at RH 65% for a certain period, mimicking a scenario where patients fail to administer the medication immediately after unpacking.

## 2. Materials and Methods

### 2.1. Materials

A lytic *Myoviridae* phage, AB406, active against *A. baumannii* was isolated and characterized by the Beijing Institute of Microbiology and Epidemiology. High titer phage lysate was obtained and titered using well-established protocols [28]. The phage lysate was then purified by anion-exchange chromatography using a CIMmultus™ QA 1 mL Monolithic Column (BIA Separations, Slovenia) [29]. The phage elution was dialyzed with phosphate-buffered saline (PBS, Sigma–Aldrich, St. Louis, MO, USA) and the obtained phage titer was 1.1 × 10^10^ pfu/mL. The host *A. baumannii* strain, MDR-AB2, was isolated from the sputum sample of a patient with pneumonia at PLA Hospital 307. The powder matrix systems were composed of different amounts of D-(+)-Trehalose dihydrate, mannitol and L-leucine (Sigma–Aldrich, USA).

### 2.2. Design of Experiment–Factorial Design

A two-factor three-level (3^2^) full factorial design (Design Expert software version 12.0.1.0) [30] was used to investigate the influence of two factors on the stabilization of AB406 phage in the dry powder form and its dispersibility. The studied factors were as follows: A—trehalose to mannitol ratio (80:0, 60:20 and 40:40); B—total solid content (20, 40 and 60 mg/mL), as shown in Table 1. The range of trehalose to mannitol ratio was selected based on our previous reports on *Pseudomonas* phage [17] that a higher mannitol content would cause significant titer loss. The range of total solid content was selected based on our preliminary data on producing powder with a size distribution fall within the inhalable range (≤5 µm) [31] using the same excipient systems without phage. Formulations were prepared randomly according to the Design Expert software, but reordered in order here to improve readability. Three separate batches of powders were produced for each formulation. The responses studied were as follows: phage viability after a 6-month storage period at room temperature and RH < 20%; fine particle fraction of phage (FPF) for dispersion performed under normal conditions (<35%) and dispersing after incubating at 65% RH for 1 h.

### 2.3. Phage Powder Preparation

An aliquot of 5 mL of the phage stock was added to 45 mL excipient solution as prepared according to Table 1. The mixtures were spray dried using a Pilotech YC-500 spray dryer (Shanghai Pilotech Instruments and Equipment, Shanghai China) coupled with a high-performance cyclonic separator from Büchi (Buchi Labortechnik AG, Flawil, Switzerland) using an open-loop setting at a drying gas flow rate of 36 m^3^/h, a liquid feed rate of 1.8 mL/min and an inlet temperature of 60 °C. The outlet temperature was ~45 °C. The produced powders were collected into scintillation vials and stored in a box with silica gel placed inside a humidity-controlled chamber (RH < 25%) at room temperature until use.

### 2.4. Particle Morphology

The morphologies of the spray-dried powders were examined using a field emission scanning electron microscope (SEM) (SU-8010, Hitachi, Tokyo, Japan) at 5 kV beam accelerating voltage. The samples were scattered on a carbon tape and sputter coated with 10 nm of gold using a Sputter Coater (Quorum Q150T ES) before imaging.

### 2.5. Particle Size Distribution

Particle size distributions of the powder formulations were measured by laser diffraction using a Laser Diffraction Particle Size Analyzer LS I3 320 (Beckman Coulter, Miami, FL, USA). Approximately 5 mg of powders were suspended in 7.5 mL of chloroform. The suspended particles were de-aggregated by sonication for five minutes. Immediately after de-aggregation, the suspension was added to the sample compartment dropwise until the optimal obscuration (8–12%) was reached. The stirrer was turned on to minimize aggregation. All measurements were done in triplicate. The size distribution was expressed by the volume median diameter (VMD), and span that defined as the difference in the particle diameters at D_10_ and D_90_ divided by the VMD (D_10_ and D_90_ are defined as the particle diameters at 10% and 90% cumulative volume, respectively).

### 2.6. Particle Crystallinity

The crystallinity of the produced phage powders and corresponding single excipient powders were examined using an X-ray diffractometer (Smartlab; Rigaku, Japan) under ambient conditions. Samples were spread on glass slides and subjected to Cu Kα radiation at 80 mA and 40 kV. The scattered intensity was collected by a detector for a 2θ range of 5–40° at an angular increment rate of 5° 2θ/min. The data were analyzed using JADE5 software (V.5.0; MaterialsData, Livermore, CA, USA).

### 2.7. Dynamic Vapor Sorption (DVS)

The moisture sorption profiles of the spray-dried phage powders were analyzed using a DVS instrument (DVS-Intrinsic, Surface Measurement Systems, London, UK). Approximately 5 mg phage powder was placed at RH 0% for vapor desorption for 4 h and the weight was used as the reference mass. The sample was then subjected to a moisture ramping cycle of 0–90% RH at a step increase of 10%. Equilibrium moisture content at each RH was defined at less than dm/dt of 0.02% per minute. The moisture sorption kinetics of the phage powders at 65% RH was also studied by measuring the mass change over a period of 24 h.

### 2.8. Thermal Analysis

The thermal properties of the phage powders were analyzed by modulated differential scanning calorimetry (mDSC) using DSC 25 equipped with a refrigerated cooling system (TA Instruments, New Castle, DE, USA) and thermogravimetric analysis (TGA) (TGA6, PerkinElmer, Waltham, MA, USA). mDSC measurement was performed on freshly prepared samples using DSC 25 equipped with a refrigerated cooling system. Approximately 4 mg of sample was weighed and sealed in a Tzero aluminum pan with a pinhole in the lid. Sample was equilibrated at −10 °C for 2 min, and then heated to 200 °C at a heating rate of 5 °C/min with modulation of ±1 °C every 60 s. Nitrogen was used as the purge gas at a flow rate of 50 mL/min. For TGA measurement each sample (5 ± 1 mg) was weighed in an alumina crucible and heated from 25 to 400 °C at a rate of 10 °C/min with dynamic nitrogen flow. The experiments were independently conducted twice.

### 2.9. In Vitro Aerosol Performance

The in vitro aerosol performance of the phage powders were evaluated according to the British Pharmacopoeia (2016) using a multistage liquid impinger (MSLI). For each dispersion, ~10 mg of phage powders were put into a size 3 hydroxypropyl methylcellulose capsule (Capsugel, NSW, Australia) in a humidity controlled chamber (<20% RH). The dispersion was performed using an Osmohaler^TM^ operated at 95 L/min for 2.5 s at room temperature and <35% RH. PBS was used as a rinsing solvent to determine the viable phage deposition profile by plaque assay. The experiment was performed in triplicate. The lower cutoff diameters of the MSLI stages 1–4 at 95 L/min are 10.33, 5.40, 2.46 and 1.35 µm, calculated with the adjustment equations given in Appendix XII C of the British Pharmacopoeia. The recovered rate was determined by the phage titer recovered from all parts (capsule, device, induction port and all the stages of the MSLI) compared with the loaded phage in the powders. The fine particle dose (FPD) of phage/sugar was defined as the phage/sugar recovered from particles with an aerodynamic diameter ≤5.0 µm. The fine particle fraction (FPF) was defined as the FPD with respect to the total recovered dose. To mimic the administration process of phage powder in high-humidity conditions, the capsulated phage powders were incubated in a high-humidity condition (65% RH) for 1 h, followed by the dispersion test described above.

### 2.10. Quantification of Trehalose by HPLC

The deposition of trehalose at the capsule, inhaler, adaptor and each part of the MSLI was determined using a UPLC system (Model Code CHA D16CHA036G; Waters, Milford, MA, USA) using RI detection. The configuration consisted of an Acquity series of quaternary solvent manager, column manager, RI detector, sample manager-FTN, and Empower^®^ software. An amino acid column (Phenomenex Luna NH_2_, 3 μm, 100 Å, 150 × 2 mm) was used. The mobile phase was a mixture of water (30%) and acetonitrile (70%). A volume of 300 μL dispersion sample was mixed with 700 μL acetonitrile before HPLC analysis. The calibration curve for trehalose was linear in the concentration range of 0.05–0.2 mg/mL (R^2^ = 0.999, *n* = 3).

### 2.11. Phage Stability in Powder

The concentrations of viable phage in the powder samples immediately after preparation and after storage were determined [32]. Phage powders were first dissolved in PBS to a final concentration of the corresponding total solid content before spray drying. Serial dilutions of the solution samples were performed by adding 20 μL sample to 180 μL PBS. A volume of 150 μL overnight cultured host bacteria was mixed with 4 mL molten soft agar (0.7% agar, 50 °C). The mixture was overlayed onto a solidified NB agar plate made of 1.5% agar (AGAR NO.1, OXOID, Hampshire, UK) and Nutrient Broth (NB, OXOID, Hampshire, UK). Then, 10 μL of diluted phage samples were dropped onto the agar lawn and left to air dry for 30 min, and the plates were incubated overnight at 37 °C. Samples that gave rise to 3–30 pfu were used to determine the phage titer. The phage stability test was performed at 20, 45 days, 2, 3 and 6 months after preparation.

### 2.12. Statistical Analysis

The analysis of each response of factorial design (stability, FPF and FPF decreased after exposure to the high-humidity condition) was performed in Design Expert software version 12.0.1.0 using Quadratic modeling. One-way analysis of variance (ANOVA) and Fisher pairwise comparison using Minitab were employed to identify any statistically significant differences in production and storage loss, and FPF of phage. A *p* value of <0.05 was considered statistically significant.

## 3. Results

### 3.1. Production Loss AB406 Phage

The viability of AB406 phage immediately after the powder preparation was determined to calculate the production loss (Figure 1). All phage powder formulations noted a titer reduction ranged 0.3–0.5 log. The results were consistent with previous studies evaluating other phages [16,33,34,35]. ANOVA analysis showed no significant differences between formulations with different compositions and total solid contents. Both trehalose–leucine and trehalose–mannitol–leucine systems could effectively protect AB406 phage from the thermal and dehydration stresses in the spray-drying process.

### 3.2. Phage and Particle Morphology

Figure 2 shows the SEM images of the spray-dried phage powders. Particles were generally spherical with a wrinkled surface. Significant particle merging was noted for formulations containing no mannitol (F1, F4 and F7) or 20% mannitol (F2, F5, and F8). On the other hand, the addition of a higher portion of mannitol (40%) to the formulations (F3, F6 and F9) could reduce the degree of particle merging. Due to the high-humidity environment in Hong Kong, exposing the produced powders to high humidity during sample collection upon powder production or during sprinkling powders to the carbon tape and golden coating procedure for SEM sample preparation for a short period of time was inevitable. Whether the particle merging was arising from the powder production process or upon the sample preparation for SEM imaging was unclear. Nonetheless, these results were consistent with our previous studies on *Pseudomonas* phage powder [16,17], even though the amount of PBS in the produced *Acinetobacter* phage powders was 10 times higher. From the SEM images, there was no difference noted between formulations prepared from different total solid contents.

### 3.3. Particle Size

The VMD and span of the particles are presented in Table 1. From the result, most particles falling within the inhalable size fraction of less than 5 µm, consistent with the SEM images. With the same total solid content, the formulation with higher mannitol content has a slightly smaller particle size (*p* < 0.05). This is possibly due to the different composition change the surface tension of the liquid feed [36,37], resulting in slightly different droplet sizes and, hence, different particles sizes. With the same excipient composition, the particle size generally increased with increasing total solid content. Although the difference is statistically significant (*p* < 0.05), the variation was very minor due to the small increase in the total solid content.

### 3.4. Powder Crystallinity

The XRD patterns of fresh phage powders (F1–F3) and spray-dried trehalose, mannitol and leucine are depicted in Figure 3. No significant differences were noted for formulations containing the same composition prepared from various total solid content (Appendix A). From the XRD result, the peak at 2θ = 6° was the primary peak for leucine which is consistent with previous study [38,39]. Except this peak, the peaks at 2θ of 18°, 24°, 29°, 30° and 32° appeared in all formulations with similar intensity, indicating the presence of crystalline leucine. Spray-dried trehalose showed a halo pattern which is a typical XRD profile for amorphous materials, confirming that the spray-dried trehalose is amorphous. The appearance of distinct peaks in the formulations of 40% mannitol are corresponding to the α-form and β-form of mannitol crystals. This was consistent with Kaialy et al. that the XRD profile of α–mannitol specific peaks at 2θ of 10°, 14° and 17° and β–mannitol specific peaks at 2θ of 11°, 15°, 23° and 30° [40].

### 3.5. Residual Moisture Content and Glass Transition Temperature

We next measured the residual moisture content and Tg of the produced powders using TGA and mDSC, respectively (Table 2). The residual moisture content was determined from the TGA profiles (Appendix A) as the mass loss between 25 °C to 100 °C where solvent evaporation occurs. The spray-dried powders had a moisture content of 2−4%, irrespective of the formulation compositions and total solid contents. From the mDSC curves (Figure 4), a glass transition temperature was noted for all nine phage powder formulations and the same excipient compositions resulted in comparable Tg. The Tg of spray-dried phage powders with 80% trehalose and 20% leucine was ~110 °C and it dropped rapidly to ~45 °C and ~15 °C as the trehalose content decreased to 60% and 40%.

### 3.6. Moisture Sorption

Figure 5A shows the moisture sorption profiles of the phage powders as the RH increased from 0 to 90% at 25 °C. The total solid content did not show a strong influence on the powder moisture sorption ability in low-humidity conditions (≤60% RH) for formulations with the same compositions. Beyond 70% RH, formulations prepared from a lower total solid content tended to have a higher moisture sorption capacity. Formulations containing 80% and 60% trehalose showed the occurrence of recrystallization between 40–60% RH, consistent with our previous study [17], but no recrystallization peak was noted for formulation containing in 40% trehalose, 40% mannitol and 20% leucine. Kinetics of water vapor sorption of F1–F9 under 65% RH at 25 °C were also investigated (Figure 5B). The data revealed that trehalose recrystallization, indicated by the overshoot of change in mass %, took place within 60 min of exposure. The maximum moisture uptake of the spray-dried powders generally increased with decreasing total solid content for formulations with the same composition. With the same total solid content, formulations with 20% mannitol had the highest moisture capacity, agreeing with our previous findings that a small portion of mannitol promotes the recrystallization of trehalose [16,17].

### 3.7. In Vitro Aerosol Performance and the Effect of High Humidity

The recovery rate and FPF based on the recovered dose of the viable phage and trehalose for F1–F9 after dispersing with an Osmohaler™ at 95 L/min for 2.5 s in the normal (23 °C, <35% RH) and high-humidity conditions (23 °C, after incubating at 65% RH for 1 h) are depicted in Figure 6 and Figure 7, respectively. The deposition profiles of recovered viable phage for all formulations were provided in Appendix A. As seen in Figure 6A and Figure 7A, the recovery rates of trehalose were in the range of 90–110%, validating the appropriateness of the dispersion experiments. However, the recovery rates of phage for all formulations were <50% for both dispersion conditions, similar to our previous reports [16,17,18]. Under the normal dispersion condition (minimum exposure to humid air), the aerosol performance of the spray-dried powders were found to be formulation composition and total solid content dependent. Higher FPF was observed in formulations prepared with higher mannitol and lower total solid contents. Among all formulations, F3 gave the higher FPF value of 37% among all formulations studied.

After exposing the phage powders to 65% RH for 1 h, the phage recovery for all formulations dropped around 8% to 10% (*p* < 0.05), with no specific trends on the impacts of formulation composition and total solid content noted. On the other hand, their impacts on the FPF were found to be composition dependent. The extent of FPF reduction in formulations containing trehalose and leucine only (F1, F4 and F7) was the smallest (*p* < 0.05) among the three compositions studied (by ~5%), followed by formulations containing 40% mannitol (F3, F6 and F9, by 15% with *p* < 0.01). A very sharp drop (>30%) occurred for formulations containing in 20% mannitol (F2, F5 and F8, with *p* < 0.01). Overall, both the FPF of trehalose and phage reduced compared with powder dispersed in normal conditions.

### 3.8. Phage Storage Stability

Since dry powder inhaler products are usually transported, stored and administrated at room temperature, sufficient storage stability of phage powder preparations at room condition is essential. The titer reduction of phage powders after storing at room temperature under desiccation was determined relative to the titer measured in the fresh powder periodically (Figure 8). After a 6-month storage period, a titer loss ranging from 0.5 to 1.8 log was noted. The stabilization of phage in the powder form depended on the formulation compositions as well as the total solid contents. Phages were gradually deactivated with time for formulations containing trehalose (80%) and leucine (20%) only (F1, F4 and F7) regardless of the total solid contents. Their titer reduction upon storage was being more profound compared with formulations containing mannitol (*p* < 0.01). Overall, formulations prepared from the lowest total solid content (20 mg/mL F1–F3) failed to stabilize the incorporated phage as continuous reduction in the phage titer was noted throughout the storage period. These findings were different from our previous observation with *Pseudomonas* phages (PEV2 and PEV40), for which only 1 log titer reduction in 12 month storage [18]. For higher total solid contents, formulations containing 40% trehalose, 40% mannitol and 20% leucine (F6 and F9) provided better storage stability of phage that no-further phage loss after 1 month and <1 log storage loss was noted. Similar findings were reported by Chang et al. that higher total solid content of excipients would minimize the phage titer reduction after spray drying [21].

### 3.9. Response Surface Methodology Analysis

Response surface methodology (RSM) with a quadratic design model was used to evaluate the relationship between the independent variables (trehalose to mannitol ratio and total solid content) and three selected responses, storage stability, FPF at normal dispersion conditions and FPF reduction after powders incubating in high-humidity conditions. The three-dimensional response surface graphs and corresponding two-dimensional contour plots generated by Design-Expert software are shown in Figure 9. The results indicated that the storage stability and FPF value of phage had linear relationships with the trehalose to mannitol ratio and total solid content. However, a nonlinear relationship between the two independent factors was presented in the FPF reduction of phage.

## 4. Discussion

Disaccharide sugars were reported to play an important role in stabilizing phage in solid-state formulations with trehalose, sucrose and lactose being the most superior sugars in protecting phage from stresses generated during solidification. Water replacement (replacing water to network with phage via hydrogen bonding to avoid phage protein aggregation) and/or vitrification (immobilizing the phage protein in a glassy matrix) are the two leading hypotheses accounting for their stabilization of proteins/phage in solid-state. Trehalose was chosen for our study because it has a slightly higher glass transition temperature (115 °C) compared with lactose (101 °C) and sucrose (60 °C) [41]. Additionally, more consistent stabilization effects on phage by trehalose were reported in the literature [16,17,18]. In contrast, discrepancies on the ability of lactose in stabilizing phage in the powder form were reported in the literature. While Chang et al. reported a lactose-leucine binary system could stabilize phage in the powder form [21], Vandenheuvel et al. demonstrated that lactose-only formulations were not able to maintain phage viability [20]. In addition, lactose may cause chemical degradation of proteins via the Maillard reaction which may not be favorable for the long-term storage of phage [42]. Previous reports showing mannitol alone was detrimental to phage [17,43], but the addition of a certain amount of mannitol to the trehalose–leucine system could significantly improve the power properties with less particle merging upon handling and showed no negative impacts on the storage stability of the phage [17]. However, it appeared to have an optimal trehalose-to-mannitol ratio to preserve phage in the powder form. Here, we studied the impacts of this ratio on the stability and aerosol performance of AB406 phage powders. The main reason for adding leucine in dry powder formulations is to improve the dispersibility of the powders, but recent studies showed that a small amount of leucine is needed to improve phage stability in spray-dried powders [21]. The surface-active L-leucine would tend to enrich at the particle surface to minimize the migration of phage to the air–liquid interface during the spray-drying process [39], protecting phage from surface inactivation [44]. In addition, the L-leucine would delay moisture sorption immediately after the powder formed, maintaining the powders in the amorphous form, which is essential for phage stabilization, upon sample collection [21]. Based on previous findings, 20% leucine would be sufficient in ensuring the stability of phage with acceptable dispersibility [21], though it was insufficient to form a continuous shell of crystal leucine on the surface [45]. Similar findings were reported for systems with higher leucine content (up to 40%) [18]. Therefore, we fixed the leucine content to 20% of the total solid content in the present study.

Figure 1 shows minimum titer reduction upon the powder production for all formulations (≤0.5 log). The data also agree with our previous findings on *Pseudomonas* phage that a portion up to 40% mannitol in the trehalose-mannitol-leucine systems did not result in more titer loss in spray drying [16,17]. On the other hand, the storage loss of AB406 phage obtained in the present study (Figure 8) were higher than the *Pseudomonas* phage noted in our previous study [18]. It is noteworthy that the mixing ratio of phage suspended in PBS to the excipient solution was 1:9 in the present study, 10 times higher than those used in our previous studies [16,17,18]. The employed higher phage suspension mixing ratio was originally aimed to increase the phage lung dose for effective treatment. However, the results suggested that a higher amount of residual salt might impair the storage stability of phage. Further investigation is needed to fully understand the effect of salts on the storage stability of phage in powder form.

To stabilize phage during storage in the dried state, it is necessary to remove enough water to immobilize phage inside a non-crystalline glassy sugar matrix. Thus, the glass transition temperature (Tg) is a key parameter for phage stability in the powder form. Recently, Chang et al. investigated the stabilization mechanisms of phages in spray-dried powders [46]. Their results showed that keeping the storage temperatures (Ts) at least ~46 °C below Tg (i.e., Tg–Ts ≥ 46 °C) will be essential for phage stabilization. Two factors, the residual moisture content and the incorporation of mannitol, were reported to affect the Tg of trehalose [47] and were investigated. According to Table 2, the residual moisture in all powder formulations were within the optimal range, 3–6%, for phage preservation identified in previous studies [48,49]. Due to the plasticizing effect of water, the Tg of amorphous solid samples were shown to decrease linearly with increasing moisture content [47,50]. However, our results showed that formulations with the same excipient compositions had comparable Tg in spite of their different residual moisture content. The disparity from previous reports might be attributed to the insensitivity of the TGA analysis in differentiating the removal of free water and bound water from the powders in determining the residual moisture. Nonetheless, the TGA analysis confirming there was a low level of water content left in the powders after the spray-drying process. Surprisingly, formulations with 40% trehalose, 40% mannitol and 20% leucine (F3, F6 and F9) had a Tg (~15 °C) lower than the storage temperature (22 °C), but they exhibited superior stabilization effect of the embedded *Acinetobacter* phage after 6 months of storage in low-humidity conditions (<20% RH) (Figure 8). Formulations with the highest trehalose content (F1, F4 and F7) had the storage temperature well below the Tg (Tg–Ts ~90 °C) showing a gradual decrease in phage titer during the storage period. These were different from the greater phage stability in systems with higher Tg as reported in Chang et al. [46]. Cleland et al. reported that high Tg disaccharides might be inefficient in preventing the incorporated proteins from unfolding during dehydration, but adding mannitol to disaccharides (trehalose and sucrose) inhibited aggregation and deamidation during storage to a greater extent than the disaccharide alone systems [51]. This may explain the better storage stability of AB406 in systems with mannitol, despite they have a lower Tg.

Both the size measurements (Table 1) and SEM images (Figure 2) of the produced phage powders showed that most of the particles of all nine formulations were well within the inhalable range. The in vitro aerosol performance of the produced powders, on the basis of the FPF values of sugars (≥50%), also confirmed that they were suitable for pulmonary delivery. Comparing with the sugar excipients, the recovery rate of viable phage was apparently lower even under the normal dispersion condition (Figure 6A). It was likely accounted by the deactivation of phage upon impaction to the inhaler walls during the dispersing process. Apart from the lower recovery of phage, the FPF of phage was also consistently lower than that of the trehalose, but the overall trends in response to the formulation compositions and total solid contents were similar (Figure 6B), suggesting the even distribution of phage within the particles. According to Figure 6B, the FPF value increased with increasing mannitol content. This was likely because the addition of mannitol help minimizing the degree of particle merging upon powder preparation, handling and, hence, dispersion. It might also be accounted by the slightly smaller particle size of the phage powders with higher mannitol content. Similarly, the smaller particle size of the phage powders prepared from the lower total amount of excipients might attributed to their slightly higher FPF. It is worth noting that the FPD of the nine formulations were in the order of 10^6^ to 10^7^ pfu which were comparable to previous studies on *Pseudomonas* phages [18,21]. The effectiveness of phage powders yielding similar lung dose in treating bacterial lung infection was confirmed in a murine model [22]. Therefore, the promising findings in the *Acinetobacter* phage powders warrant further evaluation on its therapeutic effects in vivo.

The XRD traces confirmed the produced powders were partially amorphous with the crystallinity increase with increasing mannitol content. Amorphous powders are thermodynamically unstable and have a high risk of recrystallization when exposed to moisture, inactivating the embedded phage [21]. To ensure the developed phage powder products are stable in different geographic locations all over the world, powder stability in high-humidity conditions should be considered during formulation development. It is feasible for the final phage product to be manufactured, transported and stored in relatively low-humidity conditions. However, an excellent formulation should also be user friendly for patients and tolerate high-humidity conditions for a reasonable time. To identify an optimal formulation suitable for global distribution, we also investigated the in vitro aerosol performance of phage powders after incubating at 65% RH for 1 h. As shown in DVS profiles (Figure 5), recrystallization takes place at 65% RH within 1 h, the recovery rate of phage and the number of particles able to reach deeper into the lungs (FPF) was expected to decrease after incubating in high-humidity conditions [39], and this was confirmed in Figure 7. As both the FPF of trehalose and phage dropped significantly, the recrystallization of the amorphous content might also lead to the formation of solid bridges between particles [21], making the powders more difficult to dispersed and hence a lower FPF. Overall, the total solid content showed minimum impacts on the FPF of phage, despite the moisture sorption capacity varied. However, the dispersion profiles of phage powders after incubating in high-humidity conditions strongly depend on the formulation compositions. The lung dose of phage powders (both sugars and phage) of formulations with 20% mannitol significantly reduced, while moderate reduction was noted in those contain no or 40% mannitol. These data are largely correlated with the moisture sorption kinetics of the phage powders that formulations with the presence of a small amount of mannitol promote uptake of moisture (Figure 5) and the extent of particle merging noted in the SEM images (Figure 2).

RSM has been a common approach to optimize pharmaceutical formulation [52]. However, the linear relationships noted in the storage loss and FPF in normal dispersion conditions with the studied formulation parameters suggested a higher mannitol content (≥40%) might be beneficial for AB406 phage powder preparation. According to our previous study on *Pseudomonas* phage [17], a mannitol content of 60% would cause significant titer loss compared with formulation with 40% mannitol. Therefore, we postulated the optimal mannitol fraction would lie between 40% to 60%. While further experiments are required to confirm this, F6 had 0.70 log pfu/mL titer loss after a 6-month storage period, a 34% FPF and 18% FPF reduction after incubation at 65% RH for 1 h, and is a promising formulation for further in vivo evaluation.

## 5. Conclusions

In summary, the influences of formulation composition and total solid content on the physicochemical properties, stability and dispersibility of spray-dried AB406 phage powders were studied. A better powder morphology was observed in the formulation with the higher mannitol percentage (40% mannitol irrespective of the total solid content ranged 20–60 mg/mL). Formulations containing the highest amount of mannitol provided the best storage stability of phage; there was no further phage loss after 1 month, with the higher total solid content resulting in lower storage loss (<1 log). The higher mannitol content was also showed to enhance powder dispersibility. The exposure of phage powders to a high-humidity environment for a short time was found to reduce the phage recovery rate and the FPF due to the recrystallization of amorphous content inactivating the embedded phage. Our results confirmed previous knowledge attained for phage powder preparation could be extended for the preparation of *Acinetobacter* phage with further investigation on the impact of PBS in the final phage powder needed. This is the first report evaluating the impacts of environmental humidity on the dispersion of phage powders, and our findings suggest that this is a factor that may need to be taken into consideration during formulation optimization.

## Figures and Tables

**Figure 1 pharmaceutics-13-01162-f001:**
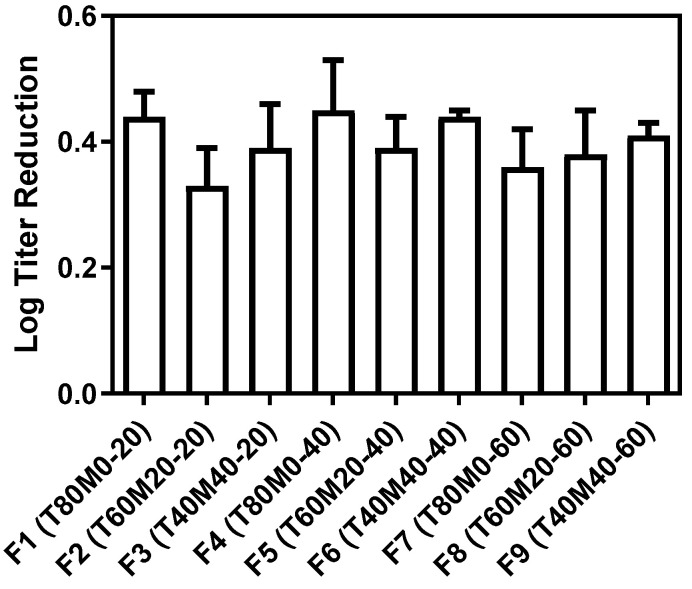
Production loss of AB406 phage in the spray-drying process. Data presented as mean ± standard deviation (*n* = 3).

**Figure 2 pharmaceutics-13-01162-f002:**
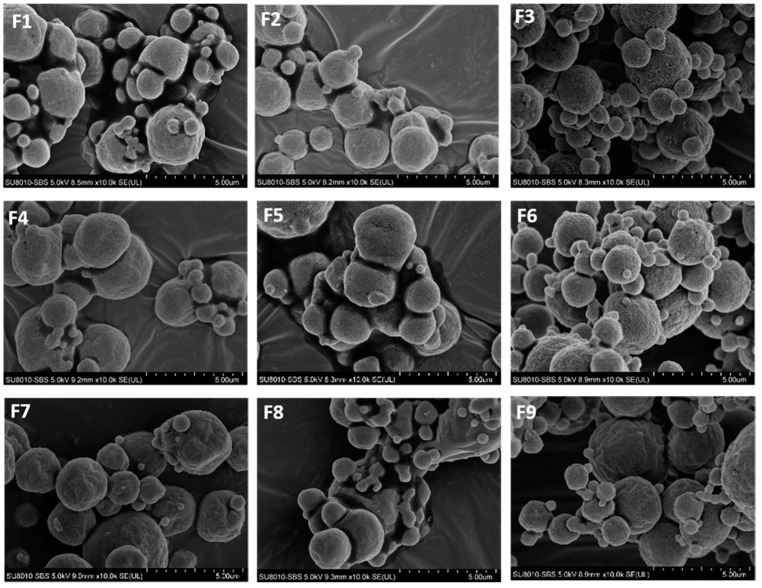
Representative SEM images of F1–F9 (5 µm scale bar).

**Figure 3 pharmaceutics-13-01162-f003:**
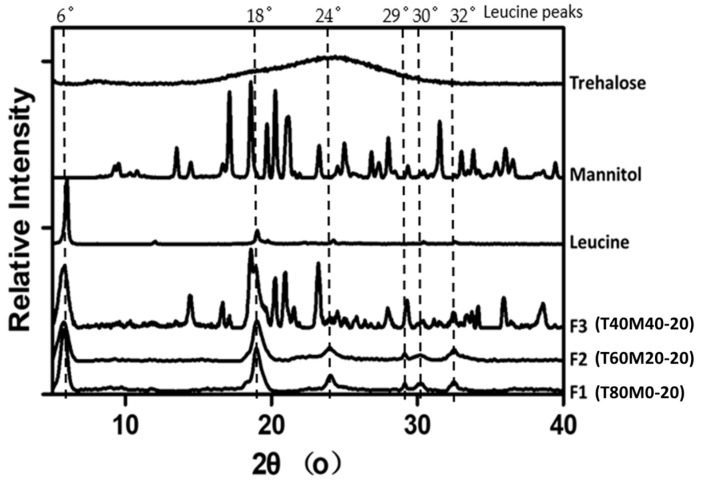
XRD profiles of the spray-dried formulations F1 to F3 and corresponding excipient.

**Figure 4 pharmaceutics-13-01162-f004:**
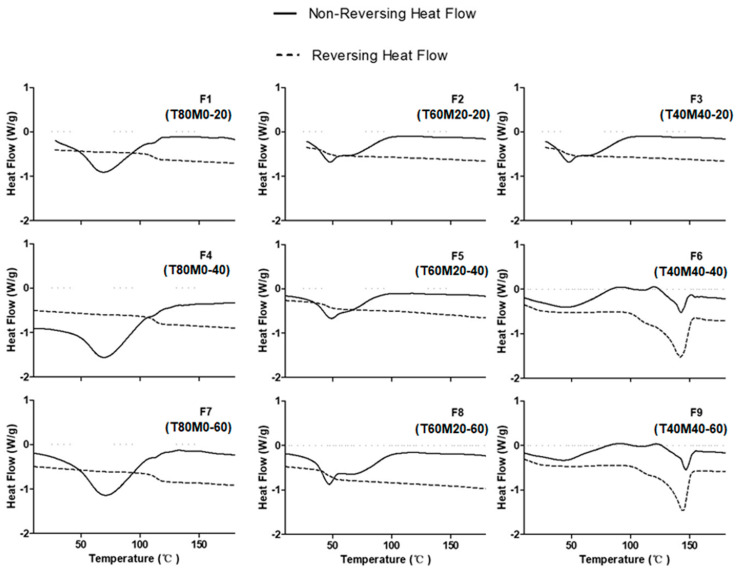
mDSC traces of the spray-dried formulations.

**Figure 5 pharmaceutics-13-01162-f005:**
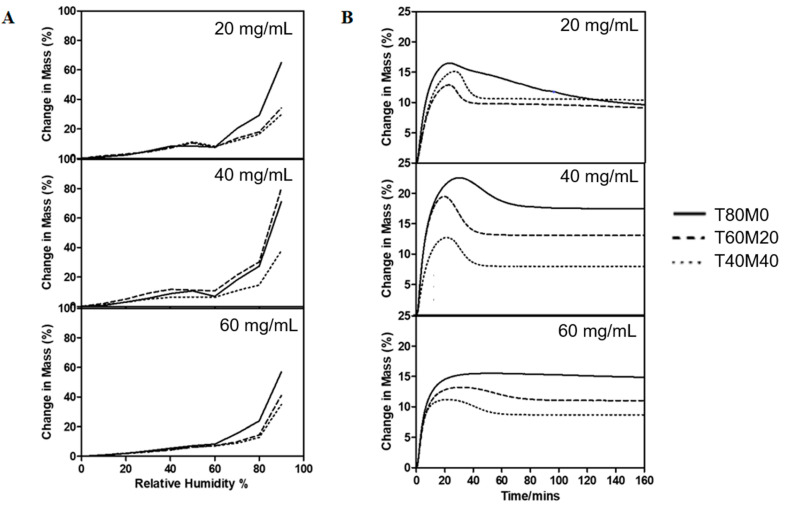
(**A**) DVS traces of F1–F9 as RH increased from 0 to 90% and (**B**) kinetics of water vapor sorption of F1–F9 under 65% RH at 25 °C.

**Figure 6 pharmaceutics-13-01162-f006:**
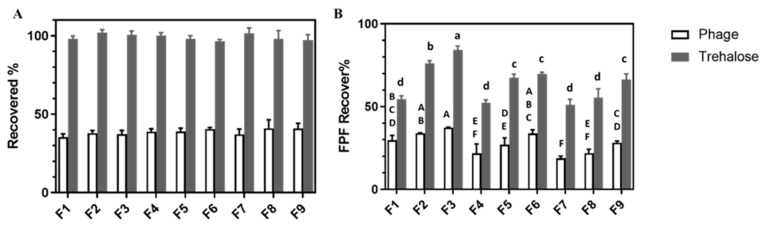
(**A**) The recovered rate and (**B**) the FPF of F1–F9. Data presented as mean ± standard deviation (*n* = 3). Column data marked with different letters indicate significant difference (*p*  <  0.05). Uppercase and lowercase letters refer to the comparison of phage and trehalose, respectively.

**Figure 7 pharmaceutics-13-01162-f007:**
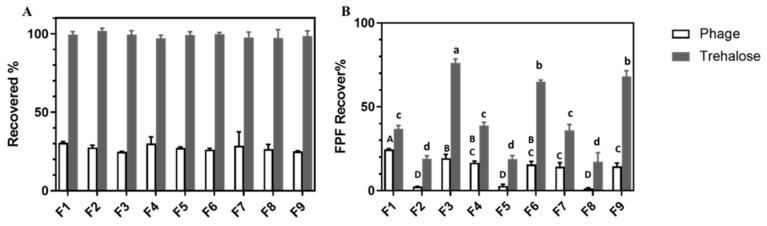
(**A**) The recovered rate and (**B**) the FPF of F1–F9 under 65% RH. Data presented as mean ± standard deviation (*n* = 3). Column data marked with different letters indicate significant difference (*p* < 0.05). Uppercase and lowercase letters refer to the comparison of phage and trehalose, respectively.

**Figure 8 pharmaceutics-13-01162-f008:**
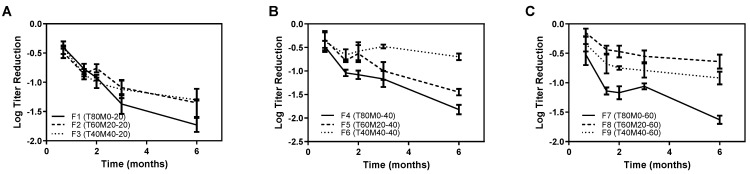
Titer reduction of phage in spray-dried powders after storage at room temperature and RH < 20% relative to titer measured in the fresh powder. (**A**) F1–F3 at a total solid content of 20 mg/mL; (**B**) F4–F6 at a total solid content of 40 mg/mL; and (**C**) F7–F9 at a total solid content of 60 mg/mL.

**Figure 9 pharmaceutics-13-01162-f009:**
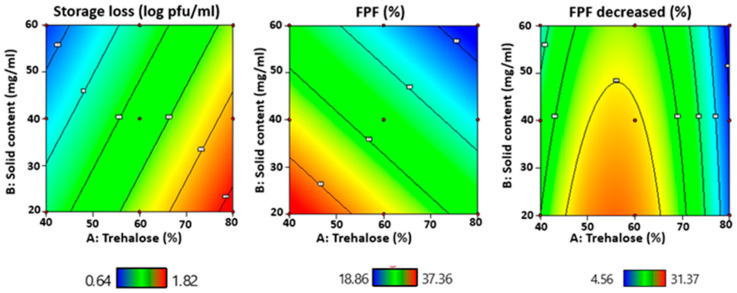
Response surface graphs depicting the influences of the percentage of trehalose and total solid content on the storage stability, FPF of phage and reduction of FPF of phage after incubating in high-humidity conditions.

**Table 1 pharmaceutics-13-01162-t001:** Formulation compositions and powder characteristics.

Formulation	Trehalose %	Mannitol %	Leucine %	Total Solid Content (mg/mL)	VMD ± SD (μm)	Span ± SD
F1 (T80M0-20)	80	0	20	20	4.59 ± 0.06	1.07 ± 0.02
F2 (T60M20-20)	60	20	20	20	3.99 ± 0.02	0.68 ± 0.04
F3 (T40M40-20)	40	40	20	20	3.69 ± 0.01	1.61 ± 0.01
F4 (T80M0-40)	80	0	20	40	4.55 ± 0.05	1.04 ± 0.01
F5(T60M20-40)	60	20	20	40	4.64 ± 0.01	1.01 ± 0.01
F6 (T40M40-40)	40	40	20	40	4.36 ± 0.01	1.02 ± 0.01
F7 (T80M0-60)	80	0	20	60	4.81 ± 0.03	1.04 ± 0.01
F8 (T60M20-60)	60	20	20	60	4.55 ± 0.01	1.01 ± 0.01
F9 (T40M40-60)	40	40	20	60	4.41 ± 0.01	1.04 ± 0.01

VMD: The volume median diameters; SD: standard deviation.

**Table 2 pharmaceutics-13-01162-t002:** Residual moisture content and glass transition temperature of F1–F9.

Formulation	Residual Moisture Content (%)	Tg (°C)
F1 (T80M0-20)	2.93	110.76
F2 (T60M20-20)	1.91	44.82
F3 (T40M40-20)	2.55	15.97
F4 (T80M0-40)	2.57	112.22
F5(T60M20-40)	2.34	45.32
F6 (T40M40-40)	1.77	17.04
F7 (T80M0-60)	3.12	112.95
F8 (T60M20-60)	3.49	45.54
F9 (T40M40-60)	2.77	16.77

## Data Availability

Not applicable.

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
