# Peer review of "The Influence of Formulation Components and Environmental Humidity on Spray-Dried Phage Powders for Treatment of Respiratory Infections Caused by Acinetobacter baumannii"

_pharmaceutics, 2021, doi:10.3390/pharmaceutics13081162_

Round 1

Reviewer 1 Report

Title: The Influence of Formulation Components and Environmental Humidity on Spray-dried Phage Powders for Treatment of Respiratory Infections Caused by Acinetobacter baumannii

Summary: The authors prepared solid formulations of a phage targeting Acinetobacter baumannii by spray-drying, utilizing trehalose, mannitol and leucine as excipients. The ratio of trehalose:mannitol and total solid content were varied. The resulting particles were characterized and the phage stability was determined after powder production, after prolonged storage at room temperature and <20% relative humidity (RH), and after 1-h incubation at 65% RH. The best conditions for phage stability out of the tested conditions were identified.

Conclusion: Accept with minor revisions.

Comments:

- Page 2 “The range of total solid content was selected based on our preliminary data on producing powder with a size distribution fall within the inhalable range using the same excipient systems without phage”: add reference for inhalable range.

- Table 1: spell out VMD and SD (under the table or in the table caption, for example).

- Page 4 “…span that defined as the difference in the particle diameters at D10 and D90 divided by the VMD.”: indicate what D10 and D90 stand for.

- Statistical analysis: although the statistical analysis was described in the methods, there are multiple parts in the text where I would be curious to know if the differences/comparisons are statistically significant and what the p-values were:

Page 6, Section 3.2 Phage storage stability: add p values for the comparisons made here

Page 7 “…the formulation with higher mannitol content has a slightly smaller particle size.”: is this statistically significant?

            Page 7 “However, the variation was very minor due to…”: is this statistically significant?

Page 10 “…the phage recovery for all formulations dropped around 8% to 10%…”: p value?

            Page 10: p value for comparisons between Fig 7b and 8b?

Page 13 “…in spite of their different residual moisture content.”: Were the differences in residual moisture content significant?

- Page 6, Section 3.2 Phage storage stability: I suggest to move this section down after the characterization of the particles, so we know what the system at hand is like, before we think about the phage stability.

- Figure 2: I would suggest to represent the x axis in scale with the time points (for example, the 2-month time point should be closer to the 1.5-month time point than to the 3-month time point. That way, the plot will give more of a sense of the trend over time. For example, the reduction we see between 3 and 6 months for F7 occurs over a course of 3 months, but it looks sharper because this 3-month period on the x-axis is as long as the other time periods.

- I would suggest to label the conditions in ways that remind the reader about the formulation of that specific sample. The current labels F1, F2 etc don’t give any information about the formulations, unless the reader keeps referring back to Table 1.

- Page 7 “Whether the particle merging was arising from the powder production process or upon the sample preparation for SEM imaging was unclear.”: please clarify you the SEM sample preparation process may have affected the morphology of the powders. If the particles were dry when applied on the carbon tape and a thin layer of gold is applied, how is the SEM sample preparation potentially affecting the morphology of the particles?

- Did you do XRD also after storage at different time points/temperature/humidity? It would be interesting for the aim of this study to see how the crystal structure changes due to those, and possibly link it with the phage stability data.

- Page 10, Section 3.8: How's the recovery rate % defined? Is it the total % of viable phages without considering the particle size (as opposed to the FPF) during the in vitro tests?

- Page 13 “Comparing with the sugar excipients, the recovery rate of viable phage was apparently lower even under the normal dispersion condition (Figure 7A). It was likely accounted by the deactivation of phage upon impaction to the inhaler walls during the dispersing process.”: It is possible that you don't recover the whole amount of phages from the surfaces although they may still be active, not just that they are deactivated during the dispersing process? Are you able to differentiate between the two different contributions?

- The text is well-written and easy to follow. Minor typos were detected, such as:

Page 2 “…but outbreak of PDR A. baumannii have been increasing reported…”: “outbreak” should be plural

Page 2 “dry powder formulations are preferred over to liquid formulations…”: delete “over”

Reviewer 2 Report

The article titled ‘The Influence of Formulation Components and Environmental Humidity on Spray-dried Phage Powders for Treatment of Respiratory Infections Caused by Acinetobacter baumanniii is well written. I have some comments before publication.

Minor English editing is required.

I suggest authors present some TEM pictures to evaluate the integrity of the phages before and after spray drying

Line 309 - Which sugar are you referring to?

Figure 7 and 8 - What do alphabets a, b, c, d (especially in Fig 7B and 8B) refer to?

Figure 8B - Why is this huge decrease in FPF? Was it because of moisture absorption? Please include explanation in discussion.

34% FPF seems low - is that enough of a dose for a therapeutic effect? The authors should include a comment in the manuscript especially since no efficacy either in vitro or in vivo were done.

Round 2

Reviewer 1 Report

I thank the authors for addressing the comments and clarifying certain points.

Reviewer 2 Report

Mentioned points have been addressed.